# Do Terrestrial Salamanders Indicate Ecosystem Changes in New England Forests?

**Ahmed A. H. Siddig** [1,2,3] **, Alison Ochs** [4,*] **and Aaron M. Ellison** [2]

1   Faculty of Forestry, University of Khartoum, Khartoum North 13314, Sudan; asiddig@eco.umass.edu
2   Harvard University, Harvard Forest, 324 N. Main Street, Petersham, MA 01366, USA;
    aellison@fas.harvard.edu
3   Department of Environmental Conservation, University of Massachusetts Amherst, 160 Holdsworth Way,
    Amherst, MA 01003, USA
4   Smithsonian Conservation Biology Institute, Conservation Ecology Center, 1500 Remount Road,
    Front Royal, VI 22630, USA
*   Correspondence: ochs22a@mtholyoke.edu; Tel.: +01-480-684-4966

**Abstract:** Long-term ecological research (LTER) and monitoring programs accrue invaluable ecological data that inform policy and improve decisions that enable adaptation to and mitigation of environmental changes. There is great interest in identifying ecological indicators that can be monitored easily and effectively to yield reliable data about environmental changes in forested ecosystems. However, the selection, use, and validity of ecological indicators to monitor in LTER programs remain challenging tasks for ecologists and conservation biologists. Across the eastern United States of America, the foundation tree species eastern hemlock (*Tsuga canadensis* (L.) Carrière) is declining and dying from irruptions of a non-native insect, the hemlock woolly adelgid (*Adelges tsugae* Annand). We use data from the Harvard Forest LTER site's Hemlock Removal Experiment together with information from other eastern US LTER sites to show that plethodontid salamanders can be reliable indicators of ongoing ecological changes in forested ecosystems in the eastern USA. These salamanders are abundant, they have a history of demographic stability, are both predators and prey, and can be sampled and monitored simply and cost-effectively. At the Harvard Forest LTER, red-backed salamanders (*Plethodon cinereus* Green) were strong indicators of intact forests dominated by eastern hemlock (*Tsuga canadensis*); their high site fidelity and habitat specificity yielded an indicator value (robust Dufrêne and Legendre's "IndVal") for this species of 0.99. Eastern red-spotted newts (*Notopthalmus viridescens viridescens* Rafinesque) were better indicators of nearby stands made up of young, mixed hardwood species, such as those which replace hemlock stands following adelgid infestation. At the Hubbard Brook and Coweeta LTER sites, plethodontid salamanders were associated with intact riparian habitats, which may also be dominated by eastern hemlock. We conclude that plethodontid salamanders satisfy most criteria for reliable ecological indicators of environmental changes in eastern US forests.

**Keywords:** ecological monitoring; foundation species; Harvard Forest; indicator species; long-term ecological research; Plethodontidae; *Tsuga canadensis*

## 1. Introduction

Long-term ecological research (LTER) refers to a systematic process of repeated, field-based empirical measurements of ecological state variables collected continuously over at least 10 years. LTER can provide temporally rich data that can identify trends and changes in environmental conditions and provide metrics of success or failure with respect to interventions intended to ameliorate

negative changes or accelerate positive ones [1–4]. Not everything can be monitored, however, and the selection and use of state variables to monitor (henceforth "ecological indicators") remains a challenging task for ecologists and conservation biologists [5–7]. Indicator species are considered to be cost-effective and intuitively appealing objects to monitor ecosystem changes because their responses to local environmental changes (e.g., changes in individual behavior or population size) are thought to integrate over a range of varying environmental conditions in their habitat [6,8,9]. Indicator species have been drawn from many taxa, including plants [10], amphibians [11–13], birds [14,15], fish [16], small mammals [17], invertebrates [18–20], and microorganisms [21,22].

The selection of indicator species and identification of variables associated with changes in the health or abundance of the indicator species need to be related directly to the context and objectives of ongoing monitoring efforts [23]. Commonly, there is a trade-off between the accuracy or precision of the data and the cost of monitoring an indicator species [6]. Numerous reviews of indicator species have suggested that the best indicators should: (1) be easy to detect and measure as well as relatively common; (2) have population sizes with small temporal variability under undisturbed conditions; (3) have known responses to natural and anthropogenic disturbances and a range of variability of these responses; (4) strongly and immediately reflect the relationships of cause and effect in ecosystem changes; (5) be predictive of management interventions; (6) integrate as many important and relatively discernible environmental features as possible (e.g., vegetation type and climate conditions) as well as be associated with unmeasured variables; and (7) be valued by human societies (e.g., [6,7,23–27]).

Several studies have suggested that amphibians, particularly plethodontid salamanders, may be used as indicator species [11,28–30]. Amphibians react quickly to changes in the environment due to their microhabitat requirements, sensitive physiologies, and dual life histories [28]. Additionally, amphibians are easily sampled and live longer than fish or invertebrates [28]. Salamanders are especially easy to measure, numerous, and sensitive to stressors [11]. Plethodontid salamanders such as the red-backed salamander (*Plethodon cinerus*) are restricted by moisture, temperature, and litter requirements that make them sensitive to environmental changes such as logging, acidification, and canopy cuts [11,29]. Furthermore, plethodontids are abundant and have a stable population size, with little interannual variability [29]. Red-backed salamanders in particular are highly abundant in the northeast; in New Hampshire, their biomass exceeded that of all other vertebrate species [11]. Therefore, plethodontids such as the red-backed salamander show the characteristics of good indicator species as they represent the status of the overall community [11], react early and predictably to ecosystem change, and are simple to examine [31].

Despite the popularity and potential of indicator species, there are several reasons to think they may not be especially effective for monitoring environmental changes. First, the responses of one or only a few populations rarely reflect the complexity of environmental change at large spatial scales, and unsampled biological interactions may have more effects on population dynamics of indicator species than the environmental variables of interest [23,32–34]. Second, the selection criteria for indicator species are often subjective, and may be influenced by unrelated goals [7,23,32,33]. Rare species may be good indicators but are often difficult to find and monitor; probability of occupancy and detection must be accounted for but may be hard to measure [33–35]. Determination of the effectiveness of an indicator species also remains an active area of research [23,36]. Finally, associations between indicator species and environmental changes of interest (i.e., monitoring goals) are often vague, qualitative, or unknown [7,23,32,35]. These associations may also change with the environment, depending on (rarely measured) genetic variations within individual populations [23,32,37].

Here, we assess the potential of two amphibian species as potential indicator species for monitoring environmental changes in eastern USA forests. Throughout its range, eastern hemlock (*Tsuga canadensis*) is being weakened and killed by a non-native insect, the hemlock woolly adelgid (*Adelges tsugae*). The loss of this late successional foundation tree species is causing a cascade of environmental and biological changes in forest stands (formerly) dominated by eastern hemlock [38,39]. We focus on two species of amphibians common in such New England hemlock

forests: red-backed salamanders (*Plethodon cinerus*; hereafter "red backs") and eastern red-spotted newts (*Notopthalmus viridescens viridescens*; hereafter "red efts"). Both species are highly abundant in the region, and their abundances have been found to decline with the loss of hemlock forest, demonstrating their potential sensitivity to changes in such systems [11,40,41]. Data are drawn from an ongoing experiment at the Harvard Forest Long-Term Ecological Research Site (HFR); some additional comparisons are made with studies of salamanders in two other forested LTER sites, the Hubbard Brook Experimental Forest (HBR) and the Coweeta Experimental Forest (CWT).

## 2. Materials and Methods

### 2.1. Study Sites

The HFR is one of 28 currently active LTER sites supported by the US National Science Foundation. The 2000-ha site is located in north-central Massachusetts (42.5319 N, −72.1915 W; Figure 1). The primary research focus of the HFR is on long-term studies of forest dynamics [42]: more information can be found at http://harvardforest.fas.harvard.edu.

The Hubbard Brook LTER is a 3076-ha site located in West Thornton, New Hampshire. Likens et al. provided a thorough discussion of its history and ecology [43]. The Coweeta LTER is a 2185-ha site located in Macon County, North Carolina. Additional information regarding its ecology and geology can be found in Elliott and Miniat [44] and Knoepp et al. [45].

### 2.2. Region-Wide Hemlock Decline and the Harvard Forest Hemlock Removal Experiment (HF-HeRE)

Eastern hemlock is a late successional foundation tree species [38] that grows in eastern North America from the state of Georgia, north into southern Canada, and west into the states of Michigan and Wisconsin (Figure 1). In many parts of its range, hemlock can account for >50% of the basal area in a given stand [46]. Hemlock stands and associated riparian habitats support unique assemblages of associated flora and fauna [30,47–56].

In the early 1950s, the hemlock woolly adelgid, native to Japan, was first detected in the eastern USA [57]. The adelgid has spread rapidly northeast and southwest from its point of introduction in Virginia, and kills >90% of hemlock seedlings, saplings, and trees in any given stand within 5–15 years of infestation [58–60]. As hemlock declines, mixed deciduous species (e.g., red maple (*Acer rubrum*) and various birch (*Betula*) species) replace it [39]. In addition to expected changes in the structure of hemlock-associated floral and faunal assemblages, the loss of hemlock is also leading to changes in ecosystem processes, including decomposition rates, nutrient availability, as well as carbon and nutrient cycling [39,61,62]. Finally, parallel changes in forest structure are occurring as people harvest hemlock before the adelgid infests a stand, so as to realize some economic return before the trees are rendered economically worthless by the adelgid [63].

The Harvard Forest Hemlock Removal Experiment (HF-HeRE) was established in 2003 to experimentally assess some of the long-term and large-scale ecological impacts of hemlock decline caused either by the adelgid or by human actions [64]. Data from HF-HeRE complement longer-term observations, the continual monitoring of hundreds of hemlock stands throughout the northeastern USA [58,65,66], and another experimental study of hemlock decline at the Coweeta LTER site [67].

HF-HeRE follows a replicated block design, with two blocks representing the ridge and valley sections of the forest. Four 0.8-ha plots were established in each block, each with a different canopy treatment to model stages of hemlock woolly adelgid infestation: hemlock control, girdling, logging, and hardwood control. Hemlock and hardwood controls represented relatively healthy hemlock-dominated forest and the end result of adelgid invasion in hardwood-dominated forest, respectively. Girdled plots had a narrow strip of bark cut from the circumference of all hemlock trees to simulate the slow death of trees infested by adelgid. Logged plots had commercially valuable trees removed to model preemptive salvage operations. Standard statistical analysis of HF-HeRE that accounts for its before-after-control-impact (BACI) design and low replication are discussed in [64].

Here, we describe the responses of two plethodontid salamanders to hemlock loss and associated ecological effects and discuss their potential to be used as indicator species for the wide range of changes in soil dynamics, arthropod community composition, and other variables observed and expected to occur in declining hemlock stands [64].

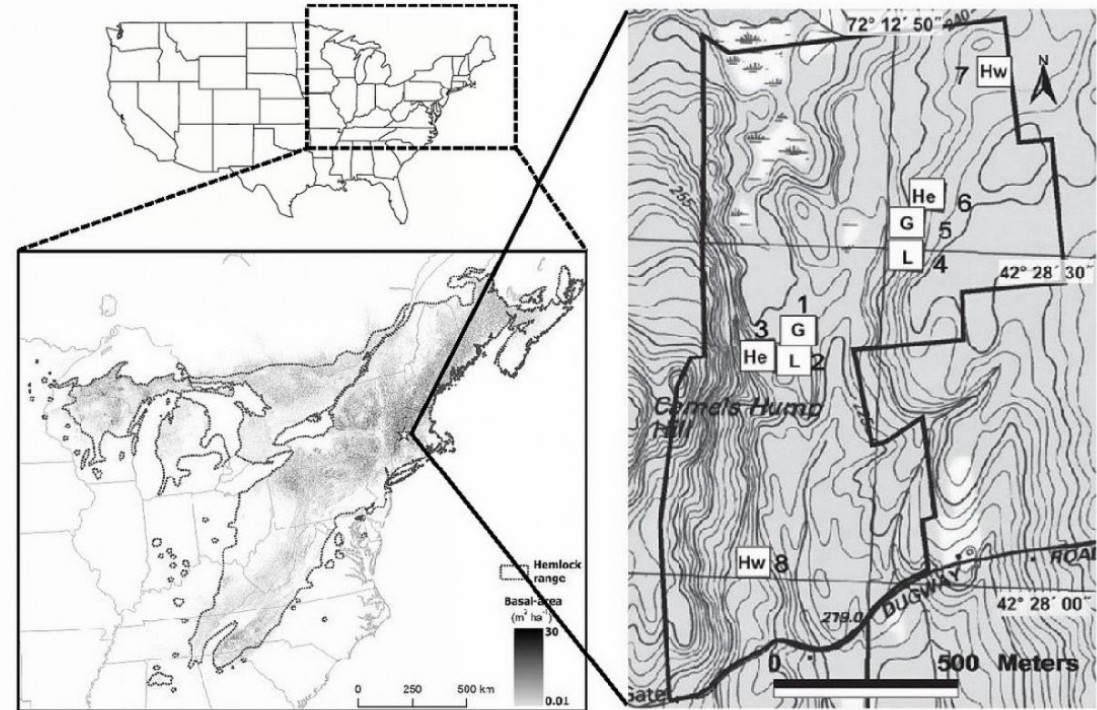

**Figure 1.** Distribution (grey shading indicating basal area in m$^2$/ha) of eastern hemlock (*Tsuga canadensis*) in eastern North America; location of the Harvard Forest (star in regional map at lower left); and layout of the Harvard Forest Hemlock Removal Experiment. In the right-hand panel, the treatments applied to each 0.81-ha experimental plot are abbreviated: He—unmanipulated hemlock control; Hw—unmanipulated hardwood control; G—bark and cambium of all hemlock seedlings, saplings, and trees girdled with chainsaws and knives; L—all merchantable hemlock ($\geq$20 cm diameter at breast height (DBH)) and some merchantable hardwoods and white pines (*Pinus strobus*) cut and removed from the site [61].

### 2.3. Sampling Plethodontid Salamanders within HF-HeRE

We monitored two species of salamanders—red backs, and red efts—within the study area of HF-HeRE between 2004 and 2014 [30,41,53]. To complement data collected in 2004 and 2005, prior to the start of HF-HeRE [30], we counted the number of red backs under artificial cover boards on weekly sampling dates during the summers (July) of 2013 and 2014. In each of the eight $\approx$90 m $\times$ 90 m (0.81-ha) HF-HeRE plots (Figure 1), five 1 m $\times$ 0.25 m $\times$ 0.02 m cover boards made from rough-sawn hemlock were deployed at 15-m intervals along each of two 75-m transects. The first and last cover board in each transect were placed 15 m from the plot edge to avoid edge effects. During the same period, we also counted individual red efts using visual encounter surveys along two 60 m $\times$ 1 m strip transects that covered the area between the first and fifth cover boards mentioned above. Sampling methods were approved by Harvard University's Institutional Animal Care and Use Committee (File 13-02-144; 2 June 2014).

*2.4. Data Analysis and Data Availability*

Salamander counts under cover boards were converted to density (individuals/m$^2$) by simple multiplication. Counts of salamanders under cover boards at these sites provide reasonable estimates of "true" density estimated by removal sampling [68].

We computed Dufrêne and Legendre's indicator species value ("IndVal"), which equals the product of the species specificity (mean abundance of species in hemlock or hardwood sites compared to all sites) and fidelity (relative frequency of occurrence of the species in hemlock or hardwood sites) for red backs and red efts in the two forest types (hemlock, young mixed deciduous hardwoods) represented within HF-HeRE. Recent researchers have pointed out that the Dufrêne and Legendre's "naïve" IndVal does not account for differences in detectability (e.g., [34,35]), so we also computed a robust IndVal that includes population size and occupancy estimates [34]. Estimates of population size for each species in each experimental treatment plot were determined using *N*-mixture models; occupancy estimates were determined using single-species, single-season occupancy models [41,69,70]. All analyses discussed were performed using the unmarked R software package [71]. Details about occupancy and detectability modeling procedure, covariates used, and model evaluation are available in Appendix 1 of Siddig et al. [41].

Following the data management policies of the US LTER network of sites, all raw data from this experiment are publicly available from the Harvard Forest data archive (http://harvardforest.fas.harvard.edu), datasets HF075 (2004, 2005 salamander data) and HF246 (2013, 2014 salamander data). Alternatively, these data may be accessed via permanent digital object identifiers (doi: 10.6073/pasta/47c731d5707857885b0aa13e0ded2991 and doi: 10.6073/pasta/9a1f20f06e6674aade200fcadf42f66e, respectively).

## 3. Results

*3.1. Salamanders Are Responsive to Ecological Changes*

In 2004, prior to the implementation of experimental manipulations in HF-HeRE, both red backs and red efts were more abundant in hemlock stands than in the young mixed deciduous stands that are replacing hemlock stands as the latter decline [30]. Both species showed immediate and substantial decreases in abundance and detectability, as well as moderate reduction in occupancy, following the experimental removal of hemlock (Table 1). The estimated abundance of red backs was four times higher in intact hemlock stands than in stands in which all hemlocks were killed in place by cambial girdling (two 0.81-ha plots) or logged and removed from the site (two plots). Further, the estimated abundance of red backs were twice as high in intact hemlock stands (two plots) than in young mixed deciduous stands (two plots). Similarly, the abundance and detectability of red efts was estimated to be twice as high in intact hemlock stands relative to the two types of manipulated hemlock stands, and 10% higher than in young mixed deciduous stands (Table 1).

**Table 1.** Relative abundance and associated standard errors of the means of red backs and red efts (individuals/m$^2$) within the two experimental treatments and two controls (two plots each) of the Harvard Forest Hemlock Removal Experiment before (2004, 2005) and after canopy manipulations simulating adelgid outbreak (girdling) or human responses to the adelgid (logging) (2013, 2014). Pre-treatment data from Mathewson [30].

| Experimental Treatment | Red Backs/m$^2$ | | | | Red Efts/m$^2$ | |
|---|---|---|---|---|---|---|
| | 2004 | 2005 | 2013 | 2014 | 2004 | 2014 |
| Hemlock control | 1.64 (0.13) | 0.2 (0.03) | 1.28 (0.01) | 1.68 (0.006) | 0.05 (0.004) | 0.04 (0.001) |
| Girdling | 2.32 (0.19) | 0.6 (0.07) | 0.28 (0.006) | 0.32 (0.003) | 0.06 (0.005) | 0.02 (0.0007) |
| Logging | 2 (0.16) | 0.12 (0.03) | 0.32 (0.007) | 0.08 (0.003) | 0.08 (0.004) | 0.02 (0.002) |
| Hardwood control | 1.16 (0.007) | 0.2 (0.03) | 0.92 (0.01) | 0.8 (0.005) | 0.04 (0.002) | 0.03 (0.003) |

*3.2. Abundant with Seasonal and Annual Stability*

Generally, salamanders are abundant in the forested ecosystems of the eastern USA and have stable populations compared to other taxa [11,40,72]. Our amphibian monitoring project at HF-HeRE found that red back populations have very stable seasonal and annual abundance consistent with previous studies (e.g., [11]) (Figure 2). This stability is a highly desirable measure for defining good indicator species, as any deviations from the norm may be considered a signal for changes in the population due to environmental disturbances rather than natural fluctuations.

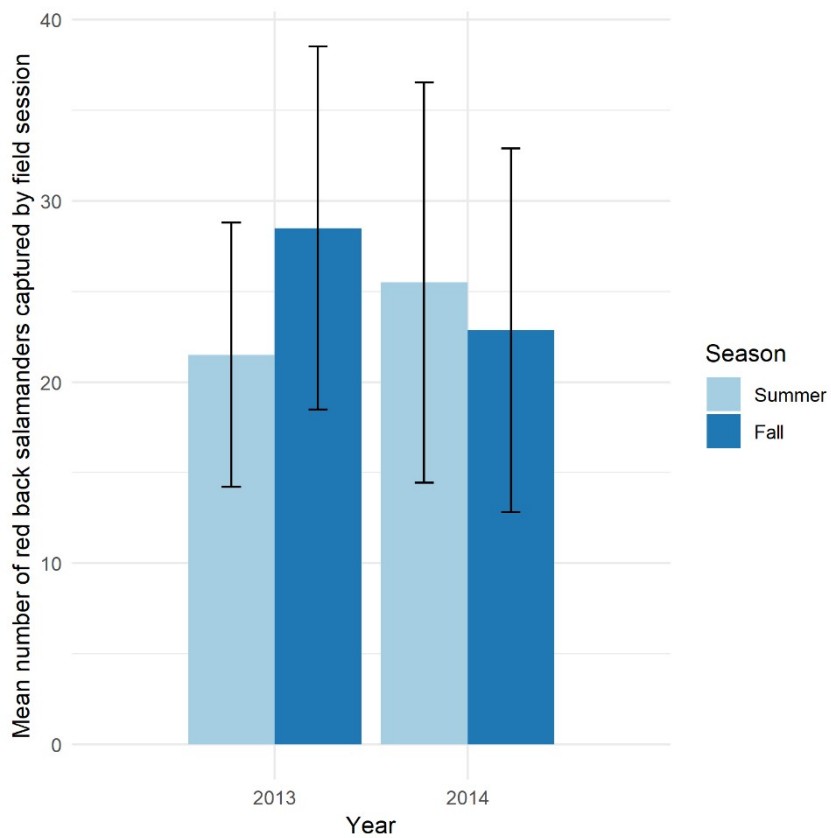

**Figure 2.** Mean seasonal and annual number of red back salamanders captured by field sessions in the Simes tract, Harvard Forest in 2013 and 2014. Lines indicate one standard error of the mean.

*3.3. Ease of Sampling and Potential for Cost-Effective Monitoring*

Both species studied are abundant on the forest floor and under natural objects. Our knowledge of their daily and seasonal activities as well as their availability for study make them readily accessible targets for sampling with cost-effective and less destructive methods such as natural or artificial object surveys [68,73]. Siddig et al. [41] found that red back detection probabilities in hemlock and mixed deciduous stands in New England forests were 0.62 and 0.25, respectively. Likewise, the detection probability for red eft populations was 0.35 in hemlock stands and 0.22 in mixed deciduous stands. Given that estimated detection probabilities can be used to predict the ability of detecting species during sampling events, we argue that both salamander species have reasonable estimates of detectability and thus reasonable ease of sampling, which supports their selection as indicators in future monitoring schemes.

*3.4. Limited Mobility and Very Local Activity*

Red backs spend a large portion of their lives underground and under natural cover within a relatively small home range (i.e., 13 m$^2$). This makes red back populations very local and thus more

likely to indicate changes in their local habitats. Likewise, red efts are normally dispersed and forage in upland habitat up to 1.5 km away from vernal pools for 3–5 years before maturing and returning to ponds to lay eggs [53]. This long residency in upland habitat with local foraging activities also makes them great predictors of changes in such habitats.

## 3.5. Indicator Values of Red Backs and Red Efts within HF-HeRE

Robust estimates revealed that both species had very high indicator values reflecting conditions in hemlock stands but relatively low indicator values for mixed deciduous forests (Table 2). Red backs had a higher IndVal for hemlock stands whereas red efts had a higher (but still low) IndVal for mixed deciduous stands.

**Table 2.** Population size, specificity, occupancy, and indicator value (IndVal) estimates for red backs (*Plethodon cinereus*) and red efts (*Notopthalmus viridescens*) in hemlock and young mixed deciduous forests in the Hemlock Removal Experiment (HF-HeRE) at the Harvard Forest long-term ecological research (LTER) site (HFR). Estimates are based on data collected in July 2014 from two 0.81-ha plots representing each forest type (HC and HD in Figure 1).

| Forest Type | Species | Estimated Population Size (Individuals Per m$^{-2}$) | Specificity | Occupancy | Naïve IndVal | Robust IndVal |
|---|---|---|---|---|---|---|
| Hemlock | Red back | 13.8 | 0.99 | 1.00 | 0.68 | 0.99 |
| | Red eft | 7.4 | 0.77 | 1.00 | 0.47 | 0.77 |
| Hardwood | Red back | 0.2 | 0.01 | 0.63 | 0.14 | 0.01 |
| | Red eft | 2.2 | 0.32 | 1.00 | 0.47 | 0.23 |

## 4. Discussion

### 4.1. The Utility of Salamanders as Indicator Species

Both salamander species that we monitored met most of the criteria for indicator species previously discussed and may be appropriate as indicator species for monitoring changes in forested ecosystems [11,41]. The rapidity of responses that we observed is supported the by findings of other research indicating that both red backs and red efts respond quickly to local environmental changes [11,74]. The substantial drop in the abundance and detectability of both species following experimental treatments within HeRE is a further indicator of their sensitivity to local habitat changes. Additionally, the large IndVal suggests that these species may be good indicators in hemlock forests, and could be used to monitor changes in ecosystem processes due to the slow decline of hemlock forests as trees are weakened by the adelgid, die, and are replaced by hardwood species.

These terrestrial salamanders are also ecologically important species in many forested ecosystems where they are abundant and centrally positioned within forest food webs (e.g., [11,40]). For example, at the Hubbard Brook Experimental Forest (HBR LTER), terrestrial salamanders were observed to account for as much biomass as small mammals, and twice the biomass of breeding birds [40]. As top-level predators of soil invertebrates that co-occur with them in the leaf litter, salamanders can affect rates of soil decomposition [72,75]; see also [76,77]. Salamanders also are prey for higher-level predators, including birds [78,79]. As plethodontid salamanders such as red backs are lungless and respire through their skin and the tissues lining their mouths, they are very sensitive to changes in air and water quality. They have small home ranges [53], long lifespans, and low temporal variability in population size [11,40,72]. Finally, they are relatively easy to sample with simple, non-destructive, and cost-effective methods such as artificial cover boards or visual encounter surveys [68,73].

### 4.2. Examples of Salamanders as Potential Indicators of Ecosystem Change at Other Forested US LTER Sites

Amphibians have been studied extensively at other LTER sites in the USA. In addition to the aforementioned work on the biomass of terrestrial plethodontid salamanders at the Hubbard Brook LTER [40], stream salamanders have been studied there for over 15 years (e.g., [80–82]). At Hubbard

Brook, Lowe [82] concluded that the abundance of *Gyrinophilus porphyriticus* was declining as the regional climate has warmed. Grant et al. [81] showed that the occupancy of four stream salamanders in northeastern forests were negatively affected by urbanization and stream configuration. Stoddard and Hayes [83] also found that several stream salamanders and other amphibian species were strongly affected by even small changes in habitat suitability, including changes to sedimentation, stream temperature, or loss of nearby vegetation due to forest management.

At the Coweeta LTER, which is near the southern range limit of eastern hemlock, Cecala et al. [84] found that riparian disturbance restricted in-stream movements of stream salamanders. These studies attest to the value of long-term monitoring at LTER sites for increasing knowledge about salamanders as well as for providing key baseline data for use in detecting long-term trends in salamander populations in response to ongoing forest disturbances, habitat loss, urbanization, and climatic change. These data could also be used to identify species that are sensitive to habitats of interest, and so could be used as indicator species.

## 5. Conclusions

Terrestrial salamanders, and especially species within the Plethodontidae, have many biological features sought after in reliable indicator species [11,23,32,41,74]. At the Harvard Forest LTER, both red backs and red efts are reasonably easy to monitor and respond quickly and sensitively to environmental changes. IndVal calculations suggested that red backs would be particularly useful as indicators of changes in declining eastern hemlock stands. As hemlocks are killed by the adelgid or removed by pre-emptive salvage logging, they are being replaced with stands of mixed hardwood species. The rate and magnitude of changes in the relative abundance of red backs and red efts as hardwoods replace hemlocks should be reliable indicators of changes in the structure and composition of ground-level and soil-inhabiting food webs associated with these two forest types. Finally, all salamanders are charismatic species that are viewed positively by non-specialists (e.g., [85]). There are additional opportunities to engage citizen scientists in monitoring salamander populations [86] that could yield data of high value for documenting environmental changes in forested ecosystems.

**Author Contributions:** Conceptualization, A.S. and A.E.; Data curation, A.S. and A.O.; Formal analysis, A.S. and A.E.; Funding acquisition, A.S. and A.E.; Methodology, A.S. and A.E.; Validation, A.S. and A.E.; Writing—original draft, A.S. and A.E.; Writing—review and editing, A.S. and A.O.

**Funding:** This research was funded by the NSF, grant numbers 0620443, 1003938, and 1237491, which support the Harvard Forest LTER and REU sites. This work was also supported by a scholarship from the Islamic Development Bank to A.A.H.S.

**Acknowledgments:** HF-HeRE is a core experiment of the Harvard Forest LTER site, and this paper is a publication of the Harvard Forest Long-Term Ecological Research Site. We thank the Harvard Forest for their support of this project and thank the staff and students who assisted with data collection and fieldwork.

**Conflicts of Interest:** The authors declare no conflict of interest.

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
