# Peer review of "Do Terrestrial Salamanders Indicate Ecosystem Changes in New England Forests?"

_forests, doi:10.3390/f10020154_

Reviewer 1 Report

Paper revision

This is an interesting and well written paper with useful suggestions for applied forest ecologists and wildlife managers. Authors performed a strong experimental design, obtaining a large set of representative data obtained with a large sampling effort. Also statistic is strong. Here, I report some suggestions and comments that, I hope, could improve the first draft of the ms.

Comments

- Row 55: Among the requisites useful to detect good indicators there are also the 'relatively commonness'. The point 1 (‘be easy to detect...’) not imply also the commonness of a species (for example, many prey birds are rare but easily detectable). Please add this point.

- Row 80-81. Regarding the subjectivity in selecting indicator species I would like to add the phenomenon for which a species is indicative only for the fact that it is actively studied by a research group that has every interest in finding a role of this type (e.g., in wildlife management).

- Row 250 and 283. ‘monitor hemlock forest decline’. It would be better to explain in what sense 'monitor the decline'. What is the factor (impacted variable) to keep under control? Size area (at patch and/or landscape level)? Habitat fragmentation and its components (e.g. isolation, distance from neighbouring patches)? degradation and edge effect? Please explain better. If 'fragmentation' is a factor, please add some references (Fahrig, Lindenmayer).

- Regarding the role as indicators of salamander, I also suggest the reading (and citation in Introduction or Discussion) of the seminal book: Villard, M. A., & Jonsson, B. G. (2009). Setting conservation targets for managed forest landscapes. Cambridge University Press. Inside this book there is and interesting chapter (Richardson JS, Thompson RM, 2009. Setting conservation targets for freshwater ecosystems in forest catchements’, pp. 244-263) where an important paper is cited: Stoddard, M. A., & Hayes, J. P. (2005). The influence of forest management on headwater stream amphibians at multiple spatial scales. Ecological applications15(3), 811-823. Here, Stoddard and Hayes reported an interesting case of salamanders acting as targets driving changes in forest management in Pacific North America.

- According to a wider point of view, among these amphibian species, is it possible to select (a priori) indicators based on their ecological traits? Yet Ewers and Didham tried to define set of species traits useful to predict (a priori) the species response to habitat fragmentation, isolation and degradation (including forests): see Ewers, R. M., & Didham, R. K. (2006). Confounding factors in the detection of species responses to habitat fragmentation. Biological reviews81(1), 117-142. I also suggest this paper for an application on reptiles: Battisti, C., & Luiselli, L. (2011). Selecting focal species in ecological network planning following an expert-based approach: Italian reptiles as a case study. Journal for nature conservation, 19(2), 126-130. Perhaps in discussion, the authors could cite as also an a-priori approach could be useful to define set of indicators for a specific taxonomic group (for amphibians this approach has not been never introduced and it could be useful to propose it for the urodel amphibians).

MINOR POINTS

In my pdf copy there are many scientific names that are not separated (name of the genus is joined to the name of the species: es.,Tsugacanadensisinstead of Tsuga canadensis). I observed this typo error in the following rows: 143, 415, 419 (caption Fig. 1). In row 449, Plethodoncinereus should be separated. See also in the row 468 for Meleagrisgallopavo. (However it is possible that this erros are only in my pdf version).

Other typographic errors:

- In caption Fig. 1 the last ‘.’ should be not separated by a white space with the last word ‘Mathewson (2009)’;

- Be careful to the white space before ‘)’ in the Table 1 (e.g. in the 2004 column: ‘(0.19 )’ should be ‘(0.19)’. See also ‘(0.16 )’ in the row below (in Table 1).

Row 297: Add a white space between ‘sites.’ and ‘This’.

In references there are authors separated by ‘&’ (e.g. row 306, 324..) and other not (e.g. rows 314, 347, 396, 474...). See Instructions for Authors of the Journal. Often ‘&’ is not separated by the name of following authors (e.g. rows 324, 360, 370, 401, 424, 478…). Check for all references.

row 352. ‘Environ.Monit.’: add a white space between words. See also ‘Ecol.Monogr.’ row 377.

row 383. add a point after ‘Front’.

Have a nice day.

Author Response

Response to Reviewer 1:

Thank you for your review.

- Row 55: Among the requisites useful to detect good indicators there are also the 'relatively commonness'. The point 1 (‘be easy to detect...’) not imply also the commonness of a species (for example, many prey birds are rare but easily detectable). Please add this point.

            This point has been added to the line.

- Row 80-81. Regarding the subjectivity in selecting indicator species I would like to add the phenomenon for which a species is indicative only for the fact that it is actively studied by a research group that has every interest in finding a role of this type (e.g., in wildlife management).

            This sentence has been extended to include this phenomenon.

- Row 250 and 283. ‘monitor hemlock forest decline’. It would be better to explain in what sense 'monitor the decline'. What is the factor (impacted variable) to keep under control? Size area (at patch and/or landscape level)? Habitat fragmentation and its components (e.g. isolation, distance from neighbouring patches)? degradation and edge effect? Please explain better. If 'fragmentation' is a factor, please add some references (Fahrig, Lindenmayer).

            This phrase has been rewritten to better explain our meaning.

- Regarding the role as indicators of salamander, I also suggest the reading (and citation in Introduction or Discussion) of the seminal book: Villard, M. A., & Jonsson, B. G. (2009). Setting conservation targets for managed forest landscapes. Cambridge University Press. Inside this book there is and interesting chapter (Richardson JS, Thompson RM, 2009. Setting conservation targets for freshwater ecosystems in forest catchements’, pp. 244-263) where an important paper is cited: Stoddard, M. A., & Hayes, J. P. (2005). The influence of forest management on headwater stream amphibians at multiple spatial scales. Ecological applications, 15(3), 811-823. Here, Stoddard and Hayes reported an interesting case of salamanders acting as targets driving changes in forest management in Pacific North America.

            Line 278-282: We include a discussion of the Stoddard and Hayes paper in the context of amphibian vulnerability to habitat changes.

- According to a wider point of view, among these amphibian species, is it possible to select (a priori) indicators based on their ecological traits? Yet Ewers and Didham tried to define set of species traits useful to predict (a priori) the species response to habitat fragmentation, isolation and degradation (including forests): see Ewers, R. M., & Didham, R. K. (2006). Confounding factors in the detection of species responses to habitat fragmentation. Biological reviews, 81(1), 117-142. I also suggest this paper for an application on reptiles: Battisti, C., & Luiselli, L. (2011). Selecting focal species in ecological network planning following an expert-based approach: Italian reptiles as a case study. Journal for nature conservation, 19(2), 126-130. Perhaps in discussion, the authors could cite as also an a-priori approach could be useful to define set of indicators for a specific taxonomic group (for amphibians this approach has not been never introduced and it could be useful to propose it for the urodel amphibians).

            Similar to other mentioned taxonomic groups, for amphibians, Welsh & Droege (11) and Welsh et al. (74) have provided a priori-selected criteria/metrics that predicting the successful use of Plethodontid salamanders as indicators of forest ecosystem changes in North America. Also recently, Siddig et al. (7), particularly in fig-6, have provided example of a priori-set of metrics that describes best indicators, including amphibians. These have been cited in our paper in the introduction and discussion.

MINOR POINTS

In my pdf copy there are many scientific names that are not separated (name of the genus is joined to the name of the species: es.,Tsugacanadensisinstead of Tsuga canadensis). I observed this typo error in the following rows: 143, 415, 419 (caption Fig. 1). In row 449, Plethodoncinereus should be separated. See also in the row 468 for Meleagrisgallopavo. (However it is possible that this erros are only in my pdf version).

            These typos have been fixed.

 Other typographic errors:

- In caption Fig. 1 the last ‘.’ should be not separated by a white space with the last word ‘Mathewson (2009)’;

- Be careful to the white space before ‘)’ in the Table 1 (e.g. in the 2004 column: ‘(0.19 )’ should be ‘(0.19)’. See also ‘(0.16 )’ in the row below (in Table 1).

Row 297: Add a white space between ‘sites.’ and ‘This’.

In references there are authors separated by ‘&’ (e.g. row 306, 324..) and other not (e.g. rows 314, 347, 396, 474...). See Instructions for Authors of the Journal. Often ‘&’ is not separated by the name of following authors (e.g. rows 324, 360, 370, 401, 424, 478…). Check for all references.

row 352. ‘Environ.Monit.’: add a white space between words. See also ‘Ecol.Monogr.’ row 377.

row 383. add a point after ‘Front’.

            All of these types have been fixed.

 Thank you very much for your comments, and I hope your concerns have been addressed.  

 Reviewer 2 Report

This manuscript makes the case for using salamanders as indicator species for forests in eastern North America. The experimental data essentially show that two salamander species respond strongly to the presence or absence of Hemlock. Presumably, rather than simply being indicators of Hemlock (for which the species itself would be a better indicator), salamanders are responding to ecosystem conditions and processes associated with intact Hemlock stands. Thus, they might be used to track decline in conditions/processes with Hemlock decline or track recovery of conditions/processes with ecosystem restoration. I think the authors could make this point more carefully and clearly in the manuscript as an indicator of Hemlock decline alone is not very useful when the abundance/density/occupancy of Hemlock itself can be (presumably) easily measured.

I should note that the experimental plot is completely unreplicated. If the blocking between ridge and valley is retained, then there is one plot for each of 8 treatments, or only 2 replicates for each of 4 treatments if the ridge/valley distinction is ignored. What this means is that the utility of salamanders as indicators beyond the Harvard Forest site is unproven. While a case for salamanders as indicators can still be made, I think that the authors need to acknowledge that they do not yet have enough evidence to support this.

I note that there is no statistical test of the significance of the IndVal measure. The usual permutation test can presumably not be done in this case because the number of possible permutations of the base data is very small (8 plots by 2 species). I really liked the approach of a robust IndVal using statistical estimates and population size and detectability.

The standard of writing is very good, although there was a couple of small errors:

line 96: should be "abundant in the region"

line 158: space missing between "efts" and "using"

Table 2: not sure what units the estimated population size is given in. It seems to be individuals per square metre - in which case, the values are far too high to be plausible.

line 267: space missing in taxonomic name.

line 276: space missing before "Also these data"

Author Response

Thank you for your review.

-This manuscript makes the case for using salamanders as indicator species for forests in eastern North America. The experimental data essentially show that two salamander species respond strongly to the presence or absence of Hemlock. Presumably, rather than simply being indicators of Hemlock (for which the species itself would be a better indicator), salamanders are responding to ecosystem conditions and processes associated with intact Hemlock stands. Thus, they might be used to track decline in conditions/processes with Hemlock decline or track recovery of conditions/processes with ecosystem restoration. I think the authors could make this point more carefully and clearly in the manuscript as an indicator of Hemlock decline alone is not very useful when the abundance/density/occupancy of Hemlock itself can be (presumably) easily measured.

            Line 139-140, and 252: A phrase has been added to emphasize systematic changes.

-I should note that the experimental plot is completely unreplicated. If the blocking between ridge and valley is retained, then there is one plot for each of 8 treatments, or only 2 replicates for each of 4 treatments if the ridge/valley distinction is ignored. What this means is that the utility of salamanders as indicators beyond the Harvard Forest site is unproven. While a case for salamanders as indicators can still be made, I think that the authors need to acknowledge that they do not yet have enough evidence to support this.

            Line 142-143: A sentence has been added with reference for clarifications regarding this issue.

-I note that there is no statistical test of the significance of the IndVal measure. The usual permutation test can presumably not be done in this case because the number of possible permutations of the base data is very small (8 plots by 2 species). I really liked the approach of a robust IndVal using statistical estimates and population size and detectability.

            Line 181-182: A reference for additional details on our statistical testing has been added.

-The standard of writing is very good, although there was a couple of small errors:

line 96: should be "abundant in the region"

line 158: space missing between "efts" and "using"

Table 2: not sure what units the estimated population size is given in. It seems to be individuals per square metre - in which case, the values are far too high to be plausible.

line 267: space missing in taxonomic name.

line 276: space missing before "Also these data"

            These typos have been fixed. We have transformed our data in to individuals per square metre and clarified the units. Both species are incredibly abundant in the region, particularly in hemlock forest (Mathewson [29]).

Thank you very much for your comments, and I hope your concerns have been addressed.

Round  2

Reviewer 2 Report

I reviewed a previous version of this manuscript. I am satisfied that my previous comments have been addressed.

Just one further thing for consideration:

I suggest that the title for section 2.1 should simply be "Study sites" because this section now describes three LTER sites rather than just the Harvard Forest site.

Author Response

Thank you for your review.

-I suggest that the title for section 2.1 should simply be "Study sites" because this section now describes three LTER sites rather than just the Harvard Forest site.

    This has been edited accordingly.

Thank you very much for your comment.

Forests EISSN 1999-4907 Published by MDPI AG, Basel, Switzerland RSS E-Mail Table of Contents Alert
Back to Top